# Unveiling Novel Avenues in mTOR-Targeted Therapeutics: Advancements in Glioblastoma Treatment

**DOI:** 10.3390/ijms241914960

**Published:** 2023-10-06

**Authors:** Shilpi Singh, Debashis Barik, Karl Lawrie, Iteeshree Mohapatra, Sujata Prasad, Afsar R. Naqvi, Amar Singh, Gatikrushna Singh

**Affiliations:** 1Department of Neurosurgery, University of Minnesota, Minneapolis, MN 55455, USA; 2Center for Computational Natural Science and Bioinformatics, International Institute of Information Technology, Hyderabad 500032, India; 3College of Saint Benedict, Saint John’s University, Collegeville, MN 56321, USA; 4Department of Veterinary and Biomedical Sciences, University of Minnesota, Saint Paul, MN 55108, USA; 5MLM Medical Laboratories, LLC, Oakdale, MN 55128, USA; 6Department of Periodontics, College of Dentistry, University of Illinois, Chicago, IL 60612, USA; 7Schulze Diabetes Institute, Department of Surgery, University of Minnesota, Minneapolis, MN 55455, USA

**Keywords:** mTOR pathway, glioblastoma, PTEN, EGFR, PDGFR, PI3K-AKT signaling, mTOR inhibitor, drug delivery

## Abstract

The mTOR signaling pathway plays a pivotal and intricate role in the pathogenesis of glioblastoma, driving tumorigenesis and proliferation. Mutations or deletions in the PTEN gene constitutively activate the mTOR pathway by expressing growth factors EGF and PDGF, which activate their respective receptor pathways (e.g., EGFR and PDGFR). The convergence of signaling pathways, such as the PI3K-AKT pathway, intensifies the effect of mTOR activity. The inhibition of mTOR has the potential to disrupt diverse oncogenic processes and improve patient outcomes. However, the complexity of the mTOR signaling, off-target effects, cytotoxicity, suboptimal pharmacokinetics, and drug resistance of the mTOR inhibitors pose ongoing challenges in effectively targeting glioblastoma. Identifying innovative treatment strategies to address these challenges is vital for advancing the field of glioblastoma therapeutics. This review discusses the potential targets of mTOR signaling and the strategies of target-specific mTOR inhibitor development, optimized drug delivery system, and the implementation of personalized treatment approaches to mitigate the complications of mTOR inhibitors. The exploration of precise mTOR-targeted therapies ultimately offers elevated therapeutic outcomes and the development of more effective strategies to combat the deadliest form of adult brain cancer and transform the landscape of glioblastoma therapy.

## 1. Introduction

Glioblastoma is a highly aggressive and resistant form of malignant brain tumor originating from glial cells. It represents the most common primary brain cancer in adults and is characterized by its rapid growth, infiltrative behavior, and tendency to recur despite intensive treatment. The tumor can appear in different regions of the brain, leading to various neurological symptoms like headaches, seizures, and cognitive decline. Its invasiveness and resistance to standard therapies pose significant challenges for treatment and prognosis, making glioblastoma one of the most lethal types of cancer [1,2].

Managing glioblastoma is exceptionally difficult due to its inherent resistance to conventional therapies. Numerous factors contribute to this resistance, rendering it one of the most challenging cancers to treat and handle effectively. Glioblastomas are highly diverse tumors consisting of a varied population of cancer cells with differing genetic and molecular characteristics. Their diffuse and infiltrative growth pattern and the extension of tumor cells into the surrounding brain tissues make complete surgical removal particularly arduous [3]. Additionally, glioblastomas contain a subpopulation of cancer stem cells known as glioma stem cells (GSCs), which can possess the ability to self-renew and contribute to tumor recurrence. These GSCs exhibit increased resilience to conventional treatments and can repair DNA damage caused by chemotherapy and radiation, leading to the emergence of drug-resistant clones and tumor recurrence [4].

Furthermore, glioblastomas are situated within the brain and are shielded by the blood–brain barrier (BBB). This protective barrier restricts the entry of numerous drugs and therapeutic agents from the bloodstream into the brain, thereby limiting their access to target tumor cells. Unlike some other cancers, glioblastoma lacks clearly defined molecular targets that can be effectively addressed with precision therapies. This lack of specific targets hampers the development of highly targeted and personalized treatments. Additionally, glioblastoma exhibits a distinct immunological tumor microenvironment and often employs immunosuppressive mechanisms that impede the immune system’s ability to target tumor cells effectively [5,6]. This evasion of the immune system further curtails the effectiveness of advanced immunotherapies. Despite extensive research endeavors, there has been limited progress in developing effective treatments for glioblastoma over the years. The challenges in glioblastoma treatment stem from its high recurrence rate and relatively poor prognosis, with a median survival of around 15 months even with aggressive therapy. Overcoming these obstacles necessitates a deeper understanding of the molecular and cellular mechanisms driving glioblastoma’s resistance and the development of innovative, individualized therapeutic approaches to enhance patient outcomes [7,8,9].

The mammalian target of rapamycin (mTOR) is a crucial signaling pathway that governs multiple cellular processes, including cell growth, proliferation, metabolism, and survival. The mTOR directly regulates translation and several transcription factors, contributing significantly to glycolysis, metabolic balance, biogenesis of mitochondria, and lysosomes, as well as lipid and protein synthesis [10,11]. In addition, studies have shown the essential role of mTOR in early embryonic development, particularly in T-cell trans-differentiation and pluripotency [12,13]. The mTOR signaling pathway plays a pivotal role in driving tumor growth and progression in glioblastoma. The dysregulation of mTOR commonly occurs in glioblastoma, and researchers have linked its activation to tumor cell survival, angiogenesis, invasion, and resistance to conventional therapies. The mTOR pathway becomes activated in response to various growth-promoting signals. Active mTOR pathway phosphorylates and regulates downstream effectors, including ribosomal protein S6 kinase (S6K) and eukaryotic initiation factor 4E-binding protein 1 (eIF4E-BP1), which are involved in protein synthesis and cell growth [14].

In glioblastoma, mTOR pathway activation is often driven by genetic alterations, such as mutations in the phosphatidylinositol 3-kinase (PI3K) pathway components, including phosphatase and tensin homolog (PTEN) and epidermal growth factor receptor (EGFR), or other signaling pathways that converge on mTOR. The overexpression of growth factor receptors platelet-derived growth factor receptor (PDGFR) and their ligands vascular endothelial growth factor (VEGF) also activate the mTOR pathway in glioblastoma. The potential of mTOR as a therapeutic target in glioblastoma lies in its central role in regulating essential cellular processes critical for tumor growth and survival. The inhibition of mTOR effectively blocks the excessive cell growth and proliferation observed in glioblastoma cells, resulting in reduced tumor growth. Also, mTOR inhibition has shown anti-angiogenic effects by suppressing the formation of new blood vessels crucial for tumor sustenance. Additionally, mTOR inhibitors have demonstrated the ability to induce autophagy, promoting tumor cell death and sensitizing glioblastoma cells to other therapeutic agents [15,16]. Several mTOR inhibitors, such as rapamycin and its analogs (rapalogs, Torin-1), have been developed and tested in preclinical and clinical studies for glioblastoma treatment. While early clinical trials have shown modest benefits, there have been concerns regarding the development of resistance to mTOR inhibitors and the necessity for combination therapies to enhance treatment efficacy [17,18]. Nevertheless, the mTOR pathway plays a central role in glioblastoma pathogenesis. Targeting mTOR remains an attractive therapeutic strategy with the potential to disrupt multiple aspects of glioblastoma growth and progression. Ongoing research aims to optimize mTOR-targeted therapies to achieve better clinical outcomes in patients with this deadliest brain cancer.

The current review comprehensively explores the role of the mTOR signaling pathway in glioblastoma, elucidating insights into its involvement in tumor growth, progression, and treatment resistance. The review aims to emphasize the molecular mechanisms underlying mTOR dysregulation in glioblastoma, including the impact of genetic alterations and upstream signaling pathways. Furthermore, it seeks to assess the potential of targeting mTOR as a therapeutic approach for glioblastoma management. The review will delve into the current status of mTOR inhibitors in preclinical and clinical studies, evaluating their efficacy and limitations. Additionally, the review will discuss emerging therapeutic strategies involved in combination therapies and personalized medicine approaches to optimize mTOR-targeted treatments. By doing so, the review intends to shed light on new therapeutic avenues and potential advancements in glioblastoma treatment through targeting the mTOR pathway.

## 2. The mTOR Signaling Pathway in Glioblastoma

mTOR is a highly conserved serine/threonine kinase that plays a central role in regulating various cellular processes. The mTOR signaling pathway integrates various extracellular and intracellular signals to control cellular behavior in response to nutrient availability (amino acids and glucose), growth factors, energy status (ATP levels), and stress signals. The mTOR signaling pathway consists of two structurally separate multiprotein complexes mTORC1 and mTORC2. Functionally, mTORC1 regulates cell proliferation known as the regulatory-associated protein of mTOR complex 1 (RAPTOR). mTOR2 activates the AKT-PKB signaling pathway and modulates cytoskeleton arrangements known as rapamycin-insensitive companion of mTOR complex 2 (RICTOR) [19,20]. Both mTOR complexes belong to the phosphatidyl inositol 3’ kinase-related kinases (PIKK) group with distinct structural domain arrangements.

Raptor is a crucial scaffold protein that binds to mTOR and is essential for mTORC1 activation. It facilitates the recruitment of substrates to the complex. Proline-rich AKT Substrate 40 kDa (PRAS40) is an inhibitory subunit of mTORC1 that phosphorylates AKT, and PRAS40 dissociates from mTORC1 for its activation. Mammalian Lethal with Sec13 Protein 8 (mLST8/GβL) stabilizes the mTORC1 complex and regulates its activity. Also, the core catalytic subunits of mTORC1 and mTORC2 have distinct compositions, including some common and unique components [21]. Rictor is a specific subunit that confers rapamycin resistance to mTORC2. It is essential for the stability and function of mTORC2. Mammalian Stress-Activated Protein Kinase-Interacting Protein 1 (mSIN1) is required for the stability and activity of mTORC2. Protein observed with Rictor (Protor) stabilizes the mTORC2 complex and contributes to its function. The activation of mTORC1 is mainly mediated by the small GTPase Ras homolog enriched in the brain (Rheb), which is positively regulated by growth factors via the PI3K-AKT signaling pathway [22]. Activated mTORC1 phosphorylates key downstream effectors, such as eIF4E-BP1 and S6K1 leading to increased protein synthesis and cell growth. Additionally, mTORC1 negatively regulates autophagy, a cellular process involved in the degradation of damaged organelles and proteins. On the other hand, mTORC2 is involved in regulating cell survival, cytoskeletal organization, and other cellular processes. It phosphorylates AKT at Ser473, promoting its activation and downstream signaling [23,24].

The dysregulation of the mTOR pathway is a hallmark of glioblastoma and plays a critical role in tumor growth and progression. Genetic alterations frequently observed in glioblastoma lead to the activation of the mTOR pathway. Among these alterations, the loss of the tumor suppressor PTEN is a common occurrence and negatively regulates the mTOR pathway (Figure 1). This loss results in the constitutive activation of mTORC1, leading to increased tumor growth [17,25]. Glioblastoma cells often exhibit the hyperactivation of the PI3K-AKT pathway, a central upstream regulator of mTORC1. The activation of the receptor tyrosine kinase EGFR in glioblastoma cells leads to increased PI3K-AKT signaling and subsequent mTORC1 activation. Additionally, excessive expression of growth factors, such as EGF, PDGF, and VEGF, in the glioblastoma microenvironment stimulates receptor tyrosine kinases and activates downstream signaling pathways, including PI3K-AKT-mTOR, promoting glioblastoma cell survival and proliferation [26].

Furthermore, glioblastoma cells demonstrate altered nutrient sensing mechanisms contributing to mTOR activation. The loss of negative regulators, such as the tuberous sclerosis complex (TSC1 and TSC2), which normally inhibits mTORC1, leads to uncontrolled mTORC1 activation in response to nutrient availability. The dysregulated mTOR signaling in glioblastoma drives various cellular processes that promote tumor growth [27,28]. mTORC1 activation enhances protein synthesis and propels cell cycle progression, increasing cell proliferation and tumor growth. mTORC1 signaling stimulates the production of pro-angiogenic factor VEGF, promoting the formation of new blood vessels to supply nutrients to the growing tumor. Additionally, mTORC1 regulates metabolic pathways involved in glucose uptake and utilization, lipid synthesis, and amino acid metabolism, providing essential nutrients for tumor growth. Moreover, mTORC1 activation promotes cell survival by inhibiting pro-apoptotic signaling pathways, thus preventing cell death, and contributing to tumor cell survival [29,30].

Dysregulated mTOR signaling sustains the self-renewal capabilities of glioma stem cells (GSCs), a subpopulation of cells with tumor-initiating properties. GSCs contribute to tumor recurrence and resistance to therapies. The dysregulation of mTOR in glioblastoma highlights its significance as a therapeutic target. Targeting the mTOR pathway has been explored as a potential treatment strategy to halt glioblastoma growth and overcome resistance to conventional therapies. The intricate interplay between the mTOR pathway and other signaling cascades in glioblastoma presents a challenging yet critical area of research aimed at developing effective and personalized therapeutic approaches for patients with this aggressive brain cancer [24,31,32].

## 3. The mTOR Inhibitor and Its Limitations

The mTOR inhibitor rapamycin and its analogs, commonly referred to as rapalogs, have been extensively explored as potential therapeutic agents for various diseases, including glioblastoma. These inhibitors primarily target the mTORC1 complex and have exhibited promise in preclinical investigations as well as early phase clinical trials. Rapalogs exhibit selective inhibition of the mTORC1 complex, with limited impact on mTORC2. Drugs like temsirolimus and everolimus directly target the mTORC1 complex and have been investigated in clinical trials as monotherapies or combined with other agents for recurrent or newly diagnosed glioblastoma. These drugs have demonstrated effectiveness in preclinical studies and early phase trials. Second-generation mTOR inhibitors AZD2014 and INK128 have shown the potential to target both mTORC1 and mTORC2 complexes and induce cell death [33,34]. Table 1 provides an overview of mTOR inhibitors, and their respective pathways targeted for therapeutic intervention.

Another next-generation mTOR inhibitor, Torin-1, is a potent and selective inhibitor targeting both mTORC1 and mTORC2 complexes, providing a more comprehensive and effective blockade of mTOR signaling. It is widely utilized in research to study the mTOR signaling pathway and its role in various cellular processes. This dual inhibition strategy offers a more potent suppression of mTOR signaling [52]. Many mTOR and PI3K inhibitors were designed and developed from PI103, and they all significantly affected glioblastoma. These compounds included KU0063794, a dual mTORC1/mTORC2 inhibitor; GDC-0941, a PI3K inhibitor; and NVP-BEZ235 [44,53,54,55]. Inhibiting autophagosome maturation combined with PI103 promoted glioma apoptosis through a Bax-dependent intrinsic mitochondrial mechanism. Similar results from other research after PI103 treatment of glioblastoma cells were seen [56,57]. A phase 1 trial for many solid tumors is currently underway for NVP-BEZ235, a dual PI3K family and mTOR inhibitor that has demonstrated effectiveness in glioblastoma. Since mTORC2 is involved in cell survival and migration processes, the partial inhibition of mTOR signaling hinders the full efficacy of rapalogs. Furthermore, rapalog treatment activates the upstream PI3K-AKT signaling pathway, creating a feedback loop that leads to the reactivation of mTORC1. This feedback mechanism contributes to developing resistance to rapalog therapy [58,59]. Moreover, prolonged use of rapalogs can result in the emergence of drug-resistant clones, leading to tumor recurrence. Regarding drug delivery, rapalogs face challenges in effectively penetrating the BBB and reaching the targeted tumor cells. The structural characteristics of large hydrophobic molecules hinder the efficient penetration of rapalogs into BBB. In addition, it exhibits complex pharmacokinetics, with variable metabolism and clearance among individuals. Alongside their therapeutic benefits, they can cause various side effects, such as immunosuppression, metabolic disturbances, and impacts on glucose homeostasis [15,17,31].

Despite these limitations, rapalogs have provided valuable insights into targeting the mTOR pathway in glioblastoma and have laid the groundwork for the development of novel mTOR inhibitors [60,61]. As standalone therapies, rapalogs may need to achieve enduring tumor control. Consequently, ongoing advanced research is investigating additional potential biomarkers and developing targeted inhibitors that can accurately identify and efficiently target mTOR signaling, aiming to enhance the efficacy of glioblastoma treatment.

## 4. Potential Biomarkers for mTOR Inhibitor Response

Biomarkers play a crucial role in the diagnosis of the disease and prognosis of the treatment response. This makes identifying and selecting biomarkers vital for selecting suitable candidates for mTOR-targeted therapies in glioblastoma. Common genetic alterations, such as tumor suppressor gene PTEN loss, can activate the mTOR signaling pathway. Glioblastoma patients with intact PTEN may respond better to mTOR inhibitors due to less hyperactivation of the pathway. The PI3K-AKT signaling, upstream of mTOR, influences its activation, suggesting that patients with hyperactive PI3K-AKT might be more responsive to mTOR inhibition [62]. Moreover, glioblastoma patients with EGFR mutations or amplifications may have increased mTOR pathway activation, making mTOR inhibitors more effective in blocking downstream signaling. The expression levels of genes within the mTOR pathway or its upstream regulators may also serve as potential biomarkers, with higher expression possibly indicating improved response to mTOR inhibitors [62]. Biomarkers linked to GSCs could predict sensitivity or resistance to mTOR inhibitors. The immune profile of glioblastoma patients also influences response to mTOR inhibitors, as mTOR signaling can affect the immune response. Biomarkers related to immune cell infiltration or immunosuppression may be relevant in this context [63].

Furthermore, specific mutations or deletions in genes within the mTOR pathway could also impact the response to mTOR inhibitors, making genetic profiling valuable for patient stratification. Given glioblastoma’s propensity to develop resistance to therapies, it becomes essential to identify biomarkers associated with resistance to mTOR inhibitors [18]. Proteomic signatures linked to the mTOR pathway might be potential biomarkers for glioblastoma prognosis. Biomarkers indicating target engagement and pathway inhibition upon mTOR inhibitor treatment could aid in assessing drug effectiveness. It is imperative that discovering reliable predictive biomarkers for mTOR inhibitors in glioblastoma is challenging due to the complexity and heterogeneity of the disease.

Despite the challenges, analysis of extracellular vesicles (EVs) sourced from plasma or cerebrospinal fluid (CSF) has shown great promise as a biomarker platform for monitoring therapeutic progress in glioblastoma patients. This study has primarily concentrated on investigating the miRNA content of these EVs, showing promising potential for glioblastoma diagnosis [64]. Clinical trials and dedicated research endeavors are indispensable for validating and establishing these biomarkers, which are crucial for tailoring personalized treatment approaches for patients with glioblastoma undergoing mTOR inhibitor therapy. Consequently, advanced research is actively exploring the potential of combination therapies involving targeted treatments and immunotherapies to augment the efficacy of mTOR inhibition in treating glioblastoma.

## 5. Strategies for Targeting mTOR in Glioblastoma

Emerging strategies for targeting mTOR more effectively in glioblastoma aim to address the limitations of current mTOR inhibitors and improve treatment outcomes. These strategies include the following: (1) Combining mTOR inhibitors with other targeted agents or conventional therapies is being explored to enhance treatment response and overcome resistance. For instance, combining mTOR inhibitors with anti-angiogenic agents like bevacizumab or immune checkpoint inhibitors aims to address multiple aspects of glioblastoma pathogenesis and improve overall therapeutic efficacy. (2) Identifying specific molecular alterations in the mTOR pathway or its upstream regulators in individual patients can guide the selection of appropriate targeted therapies. (3) Utilizing nanotechnology and extracellular vesicle-based drug delivery systems to improve BBB penetration, enhance target-specific mTOR inhibitor delivery to the brain, and minimize feedback activation and toxicity. Integrating these emerging strategies and a deeper understanding of the complex molecular networks involved in mTOR signaling holds the promise of more effective mTOR-targeted therapies for glioblastoma patients (Figure 2).

### 5.1. Combination Therapies

Combination therapies involving mTOR inhibitors have shown great promise in enhancing the efficacy of mTOR inhibition in glioblastoma and other types of cancer. Combining mTOR inhibitors with conventional chemotherapeutic agents like temozolomide has synergistic effects, leading to increased tumor cell death and improved sensitivity to chemotherapy. These combinations target multiple aspects of tumor growth and resistance mechanisms, making them more effective in combating cancer [65,66]. Additionally, combining mTOR inhibitors with anti-angiogenic agent bevacizumab (VEGF inhibitor) offers a dual benefit. This combination targets the mTOR pathway and disrupts tumor angiogenesis, leading to reduced blood supply to the tumor, inhibited tumor growth, and improved delivery of therapeutic agents to the tumor site. Since the PI3K-AKT pathway is upstream of mTOR, combining mTOR inhibitors with PI3K inhibitors provides a more comprehensive inhibition of this signaling pathway [32]. This approach is promising as it may improve tumor response and reduce the risk of feedback activation and resistance.

Combining mTOR and immune checkpoint inhibitors, such as pembrolizumab or nivolumab, can enhance the anti-tumor immune response, which unleashes the immune system’s ability to target and destroy cancer cells. Inhibitors targeting DNA damage repair pathways, such as poly(ADP-ribose) polymerase (PARP) inhibitors, have shown promise in cancer therapy [15,67,68,69]. Combining these agents with mTOR inhibitors can lead to increased DNA damage and cell death in tumor cells. Combining mTOR inhibitors with EGFR inhibitors, such as erlotinib or gefitinib, can target multiple signaling pathways and potentially overcome EGFR-driven resistance [70]. Moreover, mTOR inhibitors can induce autophagy, which promotes cancer cell survival. Combining mTOR inhibitors with autophagy inhibitors effectively blocks this mechanism, leading to enhanced therapeutic response and improved treatment outcomes [71,72,73]. These combinations offer new hope for more effective cancer treatments by targeting multiple pathways and overcoming resistance mechanisms.

### 5.2. Personalized Medicines

Personalized medicine approaches in glioblastoma treatment involve the genetic profiling of mTOR pathway components to identify specific alterations within the signaling cascade or its upstream regulators in individual patients. This genomic profiling is conducted using tumor tissues or liquid biopsies to detect mutations, deletions, or amplifications in genes related to the mTOR pathway, such as PTEN, PI3K, AKT, and other key components. Potential biomarkers are identified by analyzing the genetic profile of the tumor, which can indicate the activation status of the mTOR pathway and predict the patient’s response to treatment. Based on these genetic findings, targeted therapies are selected to specifically inhibit the identified genetic abnormalities or downstream signaling components [74,75]. This may include mTOR inhibitors, PI3K inhibitors, or other agents targeting molecular aberrations. Moreover, the tumor’s genetic profile guides the selection of combination therapies that address multiple genetic alterations within the mTOR pathway or its interactions with other signaling networks. Combining targeted therapies with different mechanisms of action allows for more comprehensive pathway inhibition (Figure 2).

Regular monitoring of treatment response via imaging, molecular analysis, and biomarker assessments helps assess the effectiveness of personalized therapies and allows for adjustments if needed. The dynamic nature of personalized medicine enables treatment adaptability based on changes in the tumor’s genetic profile or the emergence of new resistance mechanisms, optimizing treatment efficacy over time [76]. Incorporating personalized medicine approaches into clinical trials facilitates the evaluation of novel targeted therapies and combination regimens. These trials provide valuable data on treatment outcomes and contribute to developing more effective personalized therapies for glioblastoma patients. Personalized medicine offers a promising approach to tailor treatment based on each patient’s unique tumor characteristics [77,78]. This personalized approach aims to improve response rates and potentially extend survival, offering new hope in the battle against this aggressive brain tumor.

### 5.3. Nanotechnology-Based Drug Delivery

Nanotechnology-based drug delivery systems offer promising solutions to overcome the challenges of delivering mTOR inhibitors to the brain and improve their efficacy in glioblastoma treatment. Nanocarriers, such as nanoparticles, liposomes, and micelles, can be engineered to be small in size and possess surface modifications that facilitate their penetration of the BBB [79]. This facilitates precise mTOR inhibitor delivery to the brain, amplifying drug concentration at the tumor site. mTOR inhibitors, such as rapalogs, often exhibit limited stability and rapid degradation in the bloodstream. Encapsulation in nanocarriers protects these drugs from enzymatic degradation, improving stability during circulation and enabling sustained release. These nanocarriers extend mTOR inhibitor circulation by reducing clearance and preventing premature release, optimizing pharmacokinetics, increasing drug exposure, and enhancing tumor accumulation. Table 2 outlines the successful delivery and tumor-inhibiting effects of various mTOR inhibitor- encapsulated nanocarriers. While Rapamycin-loaded lipid nanoparticles have a 0.6% loading capacity, they did not exhibit synergy with x-rays in the glioblastoma model (U87MG). It selectively inhibits the mTORC1 phosphorylation and is more sensitive to mTOR inhibition in normoxia compared to hypoxia [80]. The alpha-cyano-4-hydroxycinnamic acid’s therapeutic potential was improved by encapsulating it in nanoparticles. Moreover, conjugation with cetuximab appeared to improve treatment effectiveness, particularly for U251 glioblastoma cells [81]. CPT-loaded PLGA nanoparticles show prolonged survival in the GL261 glioblastoma model, whereas PBCA and PEG-liposomes nanoparticles encapsulation increased the pharmacokinetics in rats [82,83,84].

In addition, the nanocarriers can be designed with stimuli-responsive properties that release the mTOR inhibitor in response to specific cues within the tumor microenvironment. This controlled and site-specific drug release enhances drug uptake by tumor cells while minimizing off-target effects. Nanocarriers can be functionalized by targeting ligands that recognize specific receptors overexpressed on glioblastoma cells. Active targeting facilitates receptor-mediated endocytosis and enhances drug delivery to tumor cells, reducing exposure to healthy brain tissues. Targeted delivery to the tumor site, nanocarriers can reduce systemic drug exposure, thereby minimizing off-target effects and potential toxicity in other organs.

Nanocarriers also offer the possibility of delivering multiple drugs, including mTOR inhibitors and other therapeutic agents, as part of combination therapies. It can be administered through non-invasive routes, such as intravenous injection, making it more patient-friendly and accessible. While nanotechnology-based drug delivery systems hold great promise, challenges remain, such as achieving optimal drug loading, ensuring stability during storage and manufacturing, and avoiding potential immune responses. Nonetheless, ongoing research and development efforts in this area show significant potential for enhancing the delivery and efficacy of mTOR inhibitors and improving glioblastoma treatment outcomes.

### 5.4. Extracellular Vesicle as Drug Delivery Vehicle

Extracellular vesicle (EV)-based drug delivery systems have emerged as promising and targeted approaches for treating glioblastoma. EVs are small membrane-bound vesicles released by various cell types, including tumor cells, and can transport bioactive molecules, such as proteins, lipids, and nucleic acids between cells. The unique ability of EVs to cross the BBB efficiently makes them attractive candidates for delivering therapeutic cargo to glioblastoma cells. The advantage of EV-based drug delivery is their ability to target glioblastoma cells while sparing healthy brain tissue specifically. This minimizes off-target effects and reduces systemic toxicity. The encapsulation of temozolomide (TMZ), a second-generation oral alkylating chemotherapy agent into the exosomes shown to efficiently reduce tumor resistance [103].

EVs can be engineered to carry various therapeutic agents, including small molecules, RNA-based therapies, and gene-editing tools, to target tumor cells selectively and inhibit tumor growth [104,105]. Table 3 provides an overview of engineered EVs as versatile delivery vehicles for a range of therapeutic agents. In the context of the tumor microenvironment and M1 macrophage polarization, the use of rapamycin-tumor cells-derived exosomes (texosomes) from 4T1 breast cancer cells demonstrated the upregulation of M1 markers (Irf5, Nos2, CD86) and the downregulation of M2 markers (Arg, Ym1, and CD206). This treatment also led to increased levels of M1-specific cytokines (TNF-alpha, IL-1β), decreased levels of M2-specific cytokines (IL-10, TGF-beta), elevated nitric oxide (NO) concentration, enhanced phagocytosis, and reduced efferocytosis, ultimately promoting M1 polarization. These findings suggest the potential of this approach as an immunotherapy for triple-negative breast cancer cells [106]. Additionally, Rapamycin-encapsulated U937 macrophage-derived exosomes mimic nanoparticles-in-microspheres (RNM) that were formulated with poly-lactic-co-glycolic acid (PLGA) and utilized in the treatment of hemangioma. The investigation conducted on hemangioma stem cells (HemSCs) demonstrated potent inhibition of cellular proliferation, induction of cellular apoptosis, and significant suppression of angiogenesis factor expression [107]. Furthermore, mesenchymal stem cell-derived small extracellular vesicles (sEVs) loaded with rapamycin were employed to treat autoimmune uveitis. Subconjunctival administration of rapamycin-loaded sEVs in experimental autoimmune uveitis (EAU) exhibited specific targeting of the retinal focal point, resulting in a significant reduction in the infiltration of inflammatory cells [108]. In another application, sirolimus (SIR), an mTOR inhibitor, was encapsulated within exosomes derived from fibroblasts and used in the treatment of restenosis. The outcomes revealed a notable decrease in Ki67, alpha-smooth muscle actin (α-SMA), and matrix metalloproteinase (MMP) markers in the arteries, effectively preventing restenosis [109].

Moreover, EVs derived from immune cells can be loaded with immunomodulatory agents to enhance the anti-tumor immune response, making them potential candidates for immunotherapy in glioblastoma. EVs can be further modified by adding targeting ligands on their surface to improve their specificity for glioblastoma cells. These ligands recognize specific receptors that are overexpressed on the tumor cells, promoting efficient uptake of EVs and drug release within the tumor microenvironment. This active targeting strategy enhances the therapeutic efficacy of EV-based drug delivery systems [141,142,143].

An additional advantage of EV-based drug delivery is their ability to overcome the limitations imposed by the blood—brain barrier. Being endogenous, EVs can naturally cross the BBB and deliver their cargo directly to the tumors. This facilitates improved drug penetration into the brain and enhances the accumulation of therapeutic agents at the tumor site [144,145]. Despite these promising advantages, specific challenges need to be addressed in developing EV-based drug delivery systems for glioblastoma [146]. These include the scalability of EV production, the stability of loaded cargo within EVs, and concerns about potential immunogenicity. Optimizing the engineering and targeting strategies to enhance the specificity and efficiency of drug delivery to glioblastoma cells remains an active area of research. By addressing these challenges, EV-based drug delivery systems hold great potential for transforming glioblastoma treatment and improving patient outcomes.

## 6. Preclinical and Clinical Studies

### 6.1. Preclinical Studies and Outcome

Preclinical studies investigating the efficacy of mTOR inhibitors in glioblastoma models have shown promising results. Numerous preclinical studies have demonstrated that mTOR inhibitors, particularly rapalogs like rapamycin and temsirolimus, can effectively inhibit glioblastoma tumor growth in mouse xenograft and orthotopic models. These inhibitors can induce cell cycle arrest and apoptosis in tumor cells, leading to reduced tumor size and progression. Combining mTOR inhibitors with radiation or chemotherapy can enhance the effects of radiation and chemotherapy, sensitizing tumor cells. Preclinical studies have shown that mTOR inhibitors can induce autophagy in glioblastoma cells by enhancing the cytotoxic effects of mTOR inhibitors. It has been demonstrated that new ATP-binding drugs including pyrazolopyrimidines inhibit mTOR and other PI3K family members. A particular drug in this class is PP242, an ATP-competitive mTOR inhibitor that inhibits mTORC1 and mTORC2 with significant potency and specificity [147]. Intraperitoneal (IP) administration of RapaLink-1 was compared to vehicle, rapamycin, or MLN0128 therapies, and patient-derived glioblastoma cells in xenograft models showed a significant decrease in tumor growth [148].

Combining mTOR inhibitors with other targeted agents has demonstrated enhanced anti-tumor effects and improved survival compared to single-agent treatments. Fan et al. indicated that PI103, a strong, ATP-competitive, and cell-permeable blocker of the PI3K family, exhibits powerful synergy with the EGFR inhibitor erlotinib in the reduction of glioma tumors without discernible toxicity [149]. Additionally, it has been demonstrated that doxorubicin and PI103 work in synergy against glioblastoma stem cells to boost apoptosis and decrease colony formation [54]. By decreasing the activation of AKT, boosting the expression of the pro-apoptotic proteins Bax and Caspase-3, and preventing radiation-induced DNA damage repair, NVP-BEZ235 makes glioblastoma cells more susceptible to irradiation and TMZ both in vitro and in vivo [58,150]. The DNA repair proteins PKC and ATM, which are involved in mediating resistance to ionizing radiation, were found to be inhibited by NVP-BEZ235 in glioblastoma [151]. Temozolomide and the dual PI3K/mTOR inhibitor XL-765 have been demonstrated to be effective against glioblastoma [152]. In xenograft glioma models, PKI-587, a dual PI3K/mTOR inhibitor, has been demonstrated to reduce tumor development and AKT phosphorylation [153]. A new PI3K/mTOR inhibitor (GDC-0084) dramatically inhibits the proliferation of glioblastoma cells in in vitro and effectively inhibits the growth of U87 MG glioblastoma in tumor-bearing mice via lowering AKT phosphorylation [154].

Additionally, preclinical studies focusing on mTOR inhibitors’ impact on GSCs have shown targeted suppression, leading to impaired tumor initiation, and reduced self-renewal capabilities. Sunayama et al. [155] showed that NVP-BEZ235 also decreased the expression of stem cell markers and the formation of sphere formation in cancer stem-like cells of glioblastoma. NVP-BEZ235 eliminates the tumorigenicity of glioblastoma stem-like cells while promoting cell differentiation in glial and neuronal lines. Due to its capacity to trigger autophagy, NVP-BEZ235 has also been demonstrated to make xenograft tumor models of glioblastoma more sensitive to radiation [156]. Kahn et al. showed that by inhibiting mTORC1/2, AZD2014 increases the radiosensitivity of glioma stem cells (GSCs) both in vitro and under orthotopic instances in vivo [157]. Treating genetically diverse glioblastoma cell lines with PP242 but not with rapamycin results in a dramatic and long-lasting reduction in AKT phosphorylation on serine 473, inhibiting tumor growth and invasiveness and preventing GSC proliferation [17].

Furthermore, mTOR inhibitors exhibit the ability to suppress tumor angiogenesis by targeting the vascular endothelial growth factor (VEGF) pathway, ultimately leading to decreased tumor growth and invasion. Recent studies have demonstrated that EZ235, a potent inhibitor of the PI3K-AKT-mTOR pathway, effectively downregulates VEGF expression and triggers autophagy in glioblastoma cells [158]. Importantly, preclinical studies underscore the significance of considering the molecular subtypes of glioblastoma when assessing the efficacy of mTOR inhibitors, as specific subtypes may exhibit better responses to mTOR inhibition, thus emphasizing the potential for personalized treatment approaches.

### 6.2. Ongoing and Completed Clinical Trials

Over the years, researchers have conducted numerous clinical trials to assess whether mTOR inhibitors, either as monotherapies or in combination therapies, effectively and safely treat glioblastoma. These trials have aimed to address the challenges observed with mTOR inhibitors as standalone treatments, such as resistance development and limited efficacy. Early-phase clinical trials have focused on assessing the safety and tolerability of rapamycin analogs in glioblastoma patients, aiming to establish the maximum tolerated dose and evaluate initial evidence of anti-tumor activity. Additionally, several clinical trials have explored combination therapies, combining mTOR inhibitors with other targeted agents, immunotherapies, or chemotherapy, to enhance therapeutic efficacy and overcome resistance mechanisms. Personalized medicine approaches have also been pursued, with some clinical trials selecting patients based on specific molecular characteristics, such as mTOR pathway activation or genetic alterations in mTOR-related genes. These trials have incorporated biomarker assessments, including genetic profiling and protein expression analysis, to identify potential predictive biomarkers for the response to mTOR inhibitors.

Rapamycin and its analogs including RAD001/everolimus, AP23573, and CCI-779/temsirolimus have been used in clinical trials to treat various cancers, including glioblastoma, with encouraging but complex outcomes. In preclinical and clinical investigations, researchers found that inhibitors targeting both mTORC1 and mTORC2 were more effective than Rapalogs. We eagerly await the results of a phase I study of AZD8055 in recurrent glioblastoma (NCT01316809) and advanced solid malignancies (NCT00973076). A phase I study (NCT01547546) has indicated that GDC-0084 demonstrates homogeneous distribution across the entire brain with high tumor-to-plasma and brain-to-plasma ratios [87]. To treat glioblastoma patients, NVP-BEZ235 has been enrolled in phase IIB research (NCT02430363) in conjunction with Pembrolizumab (MK-3475, a PD-1 monoclonal antibody). This combination therapy is justified by the fact that, following PD-1 suppression by MK-3475 injection, the PI3K/AKT pathway is diminished, and T cell activation is suppressed, facilitating tumor immune escape [55].

A clinical trial (phase IB) on patients with residual solid tumors, including glioblastoma, revealed the poor efficacy and unacceptable side effects of a combination of NVP-BEZ235/everolimus, including mucositis, diarrhea, nausea, fatigue, and an elevation in liver enzymes in serum [159]. A distinct phase I clinical trial demonstrated that XL765 can penetrate the BBB in individuals with glioblastoma relapse. The study also showed a reduction in effective phosphorylation of the mTOR substrate S6K and a decrease in the expression of the Ki67 proliferation marker [160]. A dual PI3K/mTOR inhibitor (PQR309) that exhibits potent inhibitory activity on Akt and phosphorylation of ribosomal protein S6. It is an ATP-competitive, BBB-permeable inhibitor. PQR309’s effectiveness, safety, pharmacodynamic, and pharmacokinetic effects are evaluated in patients with progressing glioblastoma in a non-randomized phase II research [55]. The anticancer efficacy of Pembrolizumab alone or in combination with NVP-BEZ235 and GDC-0941 is being investigated in a phase IIB clinical trial (NCT02430363) in patients with recurrent glioblastoma. In a phase II clinical trial, a combination therapeutic regimen of bevacizumab and everolimus administration was proven as efficacious as first-line therapy for glioblastoma (NCT00805961). Perifosine and temsirolimus were used in phase I/II clinical trials (NCT01051557) in recurrent HGGs to decrease the growth of murine glioblastoma independent of the presence or absence of PTEN [15,161].

It is crucial to carefully interpret the results of these clinical trials due to potential variations in the overall efficacy of mTOR inhibitors in glioblastoma. Differences in outcomes can be attributed to factors such as tumor heterogeneity, patient selection criteria, and trial design. While certain trials have reported promising results, others have indicated limited efficacy or increased toxicity. Ongoing research and meticulously planned clinical trials are essential for fully realizing the potential of mTOR inhibitors as an effective therapeutic strategy for glioblastoma patients. Table 4 lists clinical trials involving mTOR inhibitors for various glioma types.

## 7. Challenges for Targeting mTOR in Glioblastoma

Targeting mTOR in glioblastoma presents a series of obstacles that have impeded the widespread use of mTOR inhibitors as standalone therapies. Glioblastomas are characterized by substantial tumor heterogeneity, leading to variable mTOR pathway activation across different tumor subtypes. This heterogeneity results in varying responses to mTOR inhibitors, making achieving consistent and significant therapeutic effects difficult. Another challenge lies in rapalogs’ feedback loop activation of the upstream AKT signaling pathway as a compensatory mechanism. This feedback activation can reduce the efficacy of mTOR inhibition and foster drug resistance. Additionally, the crosstalk between the mTOR pathway and other signaling networks, such as the PI3K-AKT pathway and MAPK pathway, creates alternative survival pathways and resistance to mTOR inhibitors [162]. As standalone agents, mTOR inhibitors have demonstrated limited clinical activity in glioblastoma and other cancers. Efflux transporters, such as p-glycoprotein, can pump mTOR inhibitors out of the cancer cells, reducing drug accumulation and efficacy. Moreover, an additional challenge stems from the requirement for improved drug delivery to the glioblastoma site, given that the BBB impedes efficient drug penetration. Furthermore, the existence of GSC’s self-renewal abilities that contribute to therapy resistance, presents a hurdle in targeting mTOR specifically in GSCs and preventing tumor recurrence [163].

In glioblastoma, mTOR inhibitors may exhibit off-target effects and potential toxicities, particularly rapalogs and second-generation mTOR inhibitors. While these inhibitors are designed to specifically target the mTOR pathway, they can also impact other cellular processes, leading to various side effects. mTOR inhibitors can suppress the immune system, which may increase the risk of infections and compromise the body’s ability to fight against cancer cells. It can disrupt cellular metabolism, leading to metabolic changes such as hyperglycemia, dyslipidemia, and insulin resistance. mTOR inhibitors may affect blood cell production and lead to anemia, thrombocytopenia, or leukopenia, impacting the body’s ability to maintain healthy blood cell counts [164]. Some patients may experience neurological side effects, such as headaches, dizziness, or cognitive impairment. Liver toxicity can occur with mTOR inhibitors, leading to elevated liver enzyme levels and potential liver dysfunction. Glioblastoma patients receiving mTOR inhibitors may experience gastrointestinal side effects, including nausea, vomiting, and diarrhea. Skin reactions, such as rash and dermatitis, can occur as a side effect of mTOR inhibitors. It may impair the wound healing process, which can be of concern after surgical interventions. In some cases, mTOR inhibitors have been associated with interstitial lung disease or pneumonitis, a condition characterized by inflammation of the lung tissue [4,6].

## 8. Future Perspectives and New Therapeutic Approaches

### 8.1. Potential of Novel mTOR Inhibitors

The ongoing development of novel mTOR inhibitors shows immense promise in addressing the limitations and challenges of current mTOR-targeted therapies. Current research is actively exploring second-generation mTOR inhibitors that can effectively target both mTORC1 and mTORC2 complexes, providing a more comprehensive and potent blockade of mTOR signaling. These advancements aim to overcome the shortcomings observed with existing rapalog inhibitors, such as incomplete inhibition of mTORC2 and feedback activation of the PI3K-AKT pathway. Encouraging preclinical results have been observed with second-generation mTOR inhibitors like AZD2014 and INK128, demonstrating their ability to effectively target both mTORC1 and mTORC2 [165,166]. This suggests the potential for enhanced efficacy in suppressing mTOR signaling and improving treatment outcomes in glioblastoma and other cancers.

Another development area involves identifying specific vulnerabilities or dependencies in the mTOR pathway that can be targeted using novel agents. This approach aims to achieve more precise and selective mTOR inhibition, minimizing off-target effects, and enhance therapeutic responses. Furthermore, research is ongoing to explore the strategies to enhance the delivery of mTOR inhibitors to glioblastoma cells by effectively penetrating the BBB. Utilizing advanced drug delivery systems, such as nanotechnology-based formulations or extracellular vesicle-based drug delivery, holds promise in maximizing the anti-tumor effects of mTOR inhibitors within the brain.

Combinational therapies, wherein mTOR inhibitors are combined with other targeted agents, chemotherapy, or immunotherapies, are also being investigated. These combination approaches seek to synergistically enhance the effects of mTOR inhibition, potentially overcoming resistance mechanisms and improving treatment responses. In parallel, ongoing efforts are dedicated to identifying predictive biomarkers to aid in patient selection for mTOR-targeted therapies. Utilizing biomarker-based approaches can help identify patients who are more likely to respond favorably to mTOR inhibitors, leading to personalized treatment strategies and enhancing treatment efficacy.

As current research gains deeper insights into the molecular mechanisms underlying mTOR dysregulation in glioblastoma, novel therapeutic targets are being discovered. Novel mTOR inhibitors may target specific molecular abnormalities within the mTOR pathway, GSCs, or other interacting signaling pathways, further improving the precision and effectiveness of treatment. Overall, developing novel mTOR inhibitors is an active area of research, offering significant potential to revolutionize glioblastoma treatment. As these compounds progress through preclinical and clinical studies, they may bring about new opportunities to tackle glioblastoma’s challenges and improve patient outcomes in the future.

### 8.2. Targeting Specific Downstream Effectors of mTOR

Current research is increasingly focused on targeting specific downstream effectors of the mTOR pathway to enhance the efficacy of glioblastoma treatment. The research aims to overcome resistance mechanisms and improve treatment outcomes by modulating these downstream targets selectively. 4E-BP1 is a downstream effector of mTORC1 that regulates protein synthesis. It is upregulated in glioblastoma and contributes to tumor growth [167]. Targeting 4E-BP1 could suppress protein synthesis and inhibit tumor proliferation. eIF4E is another downstream target of mTORC1 that promotes cap-dependent translation and is associated with tumorigenesis [168]. Inhibiting eIF4E can disrupt protein synthesis and reduce the growth and survival of glioblastoma cells. S6K is a crucial effector of mTORC1 that regulates cell growth and survival. Inhibition of S6K has been shown to enhance the anti-tumor effects of mTOR inhibitors in glioblastoma cells [169].

On the other hand, AKT is a downstream effector of mTORC2 and is involved in cell survival and growth [170]. Combining mTOR and AKT inhibitors has shown promise in preclinical models by enhancing the therapeutic response and overcoming resistance. Hypoxia-inducible factor 1-alpha (HIF-1α) is stabilized under hypoxic conditions and promotes angiogenesis and cell survival [171]. Targeting HIF-1α in combination with mTOR inhibition may impair tumor angiogenesis and promote tumor cell death under hypoxic conditions. Crosstalk between the mTOR and MAPK/ERK pathways can contribute to resistance to mTOR inhibitors [172]. Combining mTOR inhibitors with MAPK/ERK pathway inhibitors may overcome resistance and enhance treatment efficacy. Furthermore, inhibiting mTORC2, in addition to mTORC1, may offer greater anti-tumor effects. Specific targeting of mTORC2 components is being explored to enhance the therapeutic impact in glioblastoma. As research progresses, these targeted approaches hold the promise of enhancing the clinical management of glioblastoma and potentially improving patient outcomes (Figure 3).

### 8.3. Potential of Immunotherapy in Combination with mTOR Inhibitor

The combination of mTOR inhibition with immunotherapy holds tremendous potential as a comprehensive therapeutic strategy for glioblastoma treatment. Both approaches have individually shown promise, and their synergy can offer a multifaceted and comprehensive therapeutic strategy for glioblastoma treatment. Glioblastomas create an immunosuppressive microenvironment that hampers the body’s natural immune response to tumor cells. mTOR inhibitors have demonstrated the ability to modulate the tumor microenvironment, reducing immunosuppression and potentially enhancing the efficacy of immunotherapy [173]. mTOR inhibition can upregulate major histocompatibility complex (MHC) expression and antigen presentation, making glioblastoma cells more recognizable by immune cells, such as T cells. It has also been shown to promote the function and proliferation of effector T cells [174,175]. When combined with immunotherapy, such as immune checkpoint inhibitors, this can lead to more potent anti-tumor immune responses.

Glioblastomas often overexpresses immune checkpoint molecules that suppress T-cell activity [176]. Combining mTOR inhibitors with checkpoint inhibitors can release the brakes on the immune system, enabling a more robust and sustained anti-tumor response. Additionally, mTOR inhibitors can enhance the activation and function of tumor-infiltrating lymphocytes, contributing to tumor-specific immune responses [177]. Immunotherapies, particularly chimeric antigen receptor (CAR) T cell therapy, can target glioma stem cells. Combining this approach with mTOR inhibitors may enhance the efficacy of CAR-T cells and prevent tumor recurrence. Moreover, mTOR inhibitors can counteract tumor-induced immune escape mechanisms, allowing immune cells to recognize and eliminate cancer cells effectively [178]. The combination of mTOR inhibitors and immunotherapies may also offer a way to overcome the resistance that can develop to each therapy individually, potentially restoring treatment sensitivity. However, certain challenges need to be addressed, including managing potential overlapping toxicities, identifying optimal dosing regimens, and refining patient selection criteria. The ongoing exploration and development of this combined therapeutic approach offers significant possibilities for more effective glioblastoma therapies.

### 8.4. Targeting Crosstalk of mTOR with Other Signaling Pathways

Targeting mTOR pathway crosstalk with other signaling pathways is an emerging strategy to overcome resistance in glioblastoma. Glioblastomas are complex tumors with multiple dysregulated signaling pathways, and cancer cells can activate alternative pathways to bypass mTOR inhibition and promote tumor growth. Co-targeting these interconnected pathways may enhance the treatment efficacy and prevent or reverse resistance. The PI3K-AKT pathway is a major regulator of cell survival and growth and closely interacts with the mTOR pathway. The cross-inhibition of both pathways has been explored to block cell signaling more effectively and inhibit glioblastoma growth [179,180]. Dual PI3K-mTOR inhibitors or combinations of PI3K inhibitors with mTOR inhibitors have shown promise in preclinical studies. The MAPK-ERK pathway is another critical signaling pathway that crosstalk with mTOR. The inhibition of both pathways has been investigated to target multiple resistance mechanisms and prevent adaptive signaling [65]. Combining mTOR inhibitors with MAPK-ERK pathway inhibitors may offer a more comprehensive approach to overcoming resistance. The Notch pathway is involved in glioblastoma stem cell maintenance and resistance [181]. Inhibiting Notch signaling in combination with mTOR inhibition has demonstrated synergistic effects in targeting glioblastoma stem cells and overcoming resistance [182]. Combining mTOR inhibitors with autophagy inhibitors may sensitize glioblastoma cells to mTOR inhibition and enhance treatment efficacy. Targeting EGFR signaling in combination with mTOR inhibition has been explored as a strategy to counteract EGFR-mediated resistance and improve therapeutic responses [183]. Targeting DNA repair pathways, such as the PI3K-like kinases (ATM, ATR), in combination with mTOR inhibition may enhance the cytotoxic effects of mTOR inhibitors and reduce DNA repair-mediated resistance.

Identifying specific crosstalk interactions and understanding how they contribute to resistance is crucial for designing effective combination therapies. Rational combination strategies may target multiple pathways simultaneously, leading to synergistic effects and improved therapeutic responses. However, careful consideration of potential overlapping toxicities and individual patient characteristics is necessary to ensure the safety and tolerability of these combination treatments. Exploring the crosstalk between the mTOR pathway and other signaling pathways presents a promising avenue to combat resistance and enhance the efficacy of glioblastoma treatment (Figure 3).

## 9. Conclusions

The dysregulation of the mTOR pathway is a prominent hallmark in glioblastoma, substantially contributing to various facets of tumor development and pathogenesis. Serving as a pivotal regulator of cellular growth, mTOR activation fosters glioblastoma cell proliferation, resulting in rapid tumor expansion and angiogenesis. Additionally, mTOR activation develops radiation and chemotherapy resistance, enhances DNA repair, promotes survival, and supports GSCs while fostering an immunosuppressive tumor microenvironment for immune evasion. Preclinical studies show that mTOR inhibitors effectively curb glioblastoma growth in mouse models. Combining mTOR inhibitors with anti-angiogenic or immune checkpoint inhibitors addresses multiple aspects of glioblastoma pathogenesis and improves overall therapeutic efficacy. Personalized medicine, guided by molecular profiling and mTOR pathway regulators, enhances glioblastoma treatment outcomes. However, mTOR-mediated glioblastoma treatment faces limitations, including off-target effects, drug toxicity, poor bioavailability, drug resistance, and feedback activation of alternative signaling pathways.

To overcome these limitations and improve treatment outcomes, a strategic investigation is essential in understanding the mechanisms through which glioblastoma cells develop resistance to mTOR inhibitors and activate feedback signaling. These mechanisms may include genetic alterations, pathway activation, and autophagy induction. This comprehensive approach encompasses the identification of specific biomarkers, the development of more potent and precise inhibitors targeting the mTOR pathway, the development of combination therapies that target multiple signaling networks associated with the mTOR pathway, the improvement of specific drug delivery systems to reduce the off-target effects, and the formulation of strategies to overcome drug resistance. The integration of personalized medicine approaches can assist in determining the most effective treatment strategies based on individual patient profiles, thereby maximizing the therapeutic potential of mTOR targeting in glioblastoma and improving patient outcomes. Additionally, developing novel mTOR inhibitors with improved specificity and potency holds promise to overcome resistance and enhance treatment responses in glioblastoma and other cancers.

## Figures and Tables

**Figure 1 ijms-24-14960-f001:**
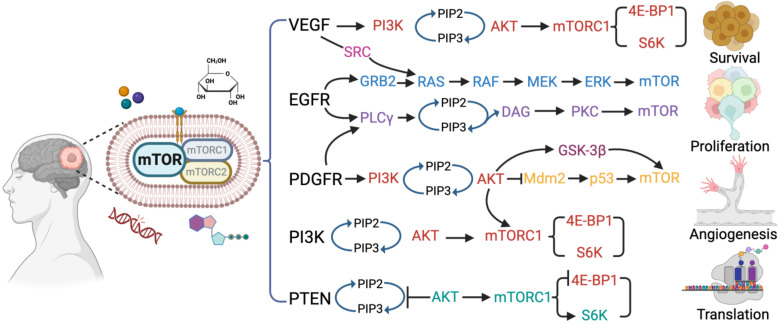
**An overview of mTOR signaling pathways and associated pathogenesis in glioblastoma.** The illustration depicting the mTOR signaling governs the cellular processes associated with the pathogenesis, such as cell growth or survival, proliferation, migration, invasion, angiogenesis, and uncontrolled protein synthesis in glioblastoma. The mTOR signaling activates the VGEF, EGFR, PDGER, PI3K-AKT, and PTEN pathways by the growth factors, genetic alteration, amino acids, and nutrients (ATP).

**Figure 2 ijms-24-14960-f002:**
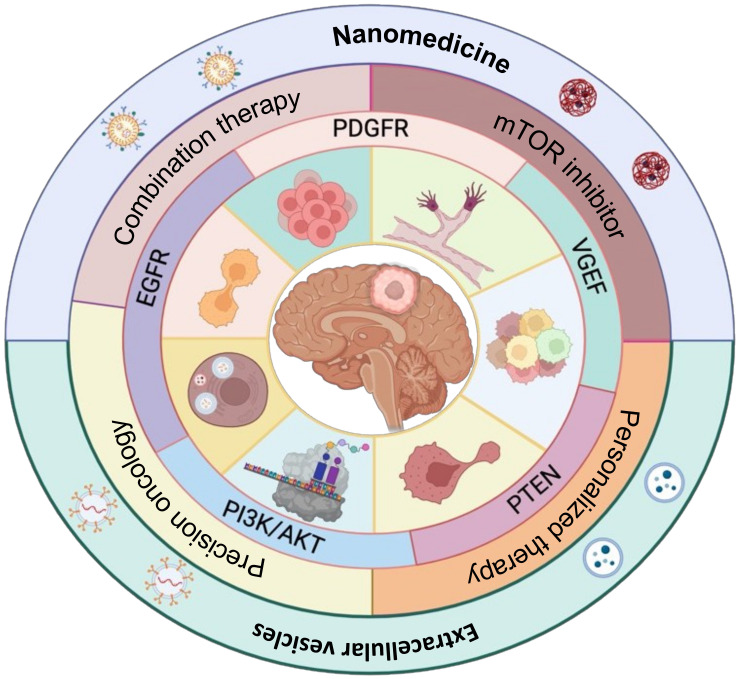
**Targeting strategies for mTOR pathways in glioblastoma.** The diagram illustrates the pathways activated via mTOR signaling and the strategies for targeting these pathways with specific therapies, including their delivery. Key pathways such as PI3K-AKT, PTEN, EGFR, VGEF, and PDGFR activate mTOR signaling, which regulates multiple cellular processes such as GSCs (Glioma Stem Cells), angiogenesis, translation, growth, migration, invasion, and autophagy in glioblastoma. The pathways targeted by mTOR inhibitors or a combination of mTOR inhibitors with advanced cancer therapies target specific delivery using nanotechnology and extracellular vesicles.

**Figure 3 ijms-24-14960-f003:**
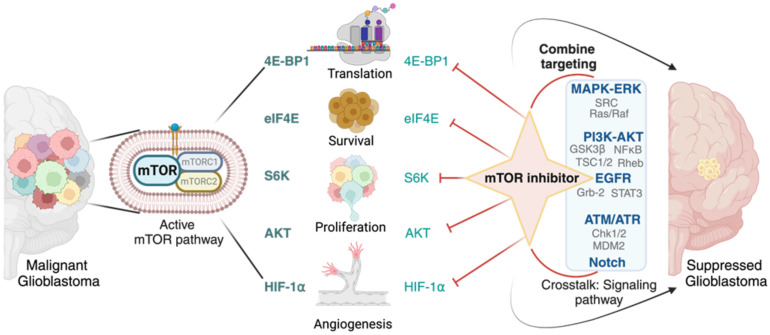
**Treatment strategy for glioblastoma therapy.** The diagram depicts the approach to target mTOR downstream effectors and interconnected signaling pathways in the treatment of glioblastoma. mTOR inhibitors can be targeted directly to the downstream effectors of mTOR signaling or applied in combination with specific inhibitors that target other pathways influenced by mTOR activation.

**Table 1 ijms-24-14960-t001:** mTOR inhibitors and their targeted pathways.

mTOR Inhibitors	Target	Activity	Limitations	References
Rapamycin	FKPB12/mTORC1	Inhibits lymphocyte activation and induces cell cycle arrest	Block mTOR activation triggers multiple feedback loops that act on upstream signaling pathways. Activation of these pathways enhances the survival and proliferation of tumor cells and induces metastasis	[35]
RAD001 (Everolimus)	mTORC1	Reduces VEGF expression and inhibits glycolysis		[36]
CCI-779 (Temsirolimus)		Inhibits mTOR activity and regulates cell division		[37,38]
AP23573 (Ridaforolimus)		Inhibits PTEN-independent tumor cell proliferation and AKT activation	Inhibition of mTOR signaling leads to disruption of normal cell functions	[39]
PP242 PP30	mTORC1 and mTORC2 in an ATP-competitive manner	Affects cell cycle, cell proliferation, and cap-dependent translation		[40]
WYE-354 WAY-600 WYE-687	mTOR/PI3K	Reduces mTORC1 and mTORC2 substrate phosphorylation in response to amino acids, and growth factors and induces PI3K-AKT	Inhibition of mTORC1 induces autophagy and promotes the tumor cell survival	[41]
Torin-1	mTOR	Inhibits mTORC1 and mTORC2 complex, impairs cell proliferation, suppresses rapamycin-resistant functions of mTORC1		[42,43]
Ku-0063794	mTOR in an ATP-competitive manner	Inhibits AKT activation and hydrophobic motif phosphorylation	Clinically significant mutations in mTOR enhance catalytic activity and reduce inhibitor efficacy	[44]
AZD8055	mTOR	Inhibits mTORC1 and mTORC2/AKT activity, promotes proteasomal degradation		[45]
XL388	mTORC1 and mTORC2	Induces apoptosis, suppresses autophagy		[46]
NVP-BEZ235 (Dactolisib)	Dual PI3K/mTOR	Inhibits AKT activity, S6RP (S6) and 4E-BP1 phosphorylation, induces FKHRL1 nuclear translocation and cell cycle arrest		[47]
PQR309 (Bimiralisib)	Dual PI3K/mTOR	Inhibits proliferation, and induces apoptosis and G1 cell cycle arrest		[48]
PKI587 (Gedatolisib)	Dual PI3K/mTOR	Increases DNA damage		[49]
JR-AB2-011	mTORC-2	Inhibits rictor-mTOR association, reduces migration and invasion		[50]
RapaLink-1	mTORC1 and mTORC2	Reduces chemotherapeutic drug resistance		[51]

**Table 2 ijms-24-14960-t002:** mTOR inhibitor-encapsulated nanocarriers as therapeutic agents and effects on tumor growth.

Drugs	Nanocarrier	Particle Size (nm)	Entrapment Efficiency (%)	Effects	References
Rapamycin	α,β-Poly(*N*-2-hydroxyethyl)-D,L-aspartamide (PHEA)-g-RhB	100	82	Efficient release and protection of Rapamycin from degradation	[85]
PI-103	Supramolecular polysaccharide nanotheranostic (SPN)	200	-	Kinase inhibition and caspase-mediated apoptosis	[86]
Rapamycin	Biomimetic nanoparticle (Leukosome)	108	-	Decrease macrophage proliferation and proinflammatory cytokines	[87]
Rapamycin	Lipid nanocapsule (LNCs)	110	69	mTORC1 signaling inhibition	[80]
Rapamycin	P80-Par-PLGA-NPs or P80-CLD-PLGA-NPs	110	69	Anti-glioma activity	[88]
Rapamycin	NP-conjugated pericardial patche	370	86	Reduction of smooth muscle cell proliferation	[89]
Rapamycin	PLGA-LTTHYKL peptide	122–130	88–91	Phospho-S6 Inhibition	[90]
Sirolimus	Cholesterol-PEG-NH2 or Cholesterol-PEG-amine	12–14	77–82	Scleral permeation and retention	[91]
Sirolimus	D-α-tocopheryl polyethylene glycol succinate (TPGS)	11	97	Improve oral absorption	[92]
Rapamycin	O-octanoyl-chitosan-polyethylene glycol (OChiPEG)	44	86	Scleral permeation and retention	[93]
Rapamycin	Pluronic block copolymer	-	-	Increase solubility and oral administration, enhance absorption	[94]
Sirolimus	Polymeric nanoparticle (PNP)	35–38	-	Increase radiosensitivity	[95]
Rapamycin +Tacrolimus	Poly(ethylene glycol)-poly(pro-pylene sulfide) (PEG-PPS)	39	41	Allograft survival	[96]
Rapamycin	Poly(ethylene glycol)-block-poly(2-methyl-2-benzoxycar-bonyl-propylene carbonate) (PEG-b-PBC)	66–76	15–88	Toxicity reduction	[97]
Rapamycin	PLGA	180	88	Reduce neointimal hyperplasia	[98]
Rapamycin + Paclitaxel	Methoxyl-poly(ethylene glycol)-succinic acid	56–94	-	Reduce multi-drug resistance	[99]
Rapamycin + Piperine	PLGA	150	-	Improve oral bioavailability	[100]
Rapamycin + Cisplatin	PLGA	12–75	93	Alteration of tumor microenvironment	[101]
17-AAG + Paclitaxel	PEG-PLA	37–44	-	Increase apoptosis	[102]

**Table 3 ijms-24-14960-t003:** Extracellular vesicles as delivery vehicles for therapeutic agents and their effects.

Therapy Category	Therapeutic Class	Therapeutic Agent	EV Source	EV Type	Packaging Method	Effects	References
Small molecules	mTOR inhibitor	Rapamycin	MSC	Small EV	Ultra sonication	Autoimmune response inhibition	[108]
		Sirolimus	Fibroblast	Exosome	Electroporation and ultra sonication	Arterial restenosis inhibition	[109]
		Rapamycin	Macrophage	Exosome	Extrusion (EB-AM)	Reduce proliferation and induce apoptosis	[107]
		Rapamycin	4T1-breast cancer	Exosome	Co-culture of 4T1-with Rapamycin	Increase M1 marker and decrease M2 marker expression	[106]
	Chemotherapy	Doxorubicin	MSC	Exosome	Electroporation	Tumor growth inhibition	[110]
		Paclitaxel	PC-3	Exosome	Co-culture	Induce cytotoxicity	[111]
		Cisplantin	Macrophage	Exosome	Co-culture	Tumor growth inhibition	[112]
		Curcumin	PANC-1	Exosome	Co-culture	Induce apoptosis	[113]
		TMZ	Glioma cell	Exosome	Co-culture	Tumor growth inhibition	[103]
		Camptothecin	4T1	Apoptotic body	Co-culture	Tumor growth inhibition	[114]
	Kinase inhibitor	TGFβRI	FBS	Exosome	Electroporation	Tumor growth inhibition	[115]
	Immune inhibitor	TLR7/8	FBS	Exosome	Electroporation	Tumor growth inhibition	[115]
		Lapatinib	MCF10A	Exosome	Electroporation	T cell activation	[116]
		CpG	EL4	Apoptotic body	Co-culture	Prevent tumor metastasis and recurrence	[117]
		cGAMP	Breast cancer cell	Apoptotic body	Active loading	STING activation and antigen representation	[118]
Antibodies		A33Ab	LIM125	Exosome	Co-culture	Tumor targeting	[119]
		MHC, CD86, αCD3, αEGFR	DC	Exosome	Co-culture	Tumor growth inhibition	[120]
		CD3, CD28	HEK293T	Exosome	Transfection	T cell activation	[121]
RNA	miRNA	miR-138-5p	ADSCs	Exosome	Transduction	Tumor growth inhibition	[122]
		miR-497	HEK293T	Exosome	Transfection	Tumor growth inhibition	[123]
		miR-199a	AMSC	Exosome	Transduction	Doxycycline sensitivity	[124]
		miR-146b	MSC	Exosome	Electroporation	Tumor growth inhibition	[125]
		miR-21	HEK293T	Exosome	Electroporation	Tumor growth inhibition	[126]
	siRNA	siS100A4	Breast cancer cell	Exosome	Co-culture and Extrusion	Tumor growth inhibition	[127]
		siSTAT3	RAW	Exosome	Ultra-sonication	Tumor growth inhibition	[128]
		siCDK1	Sk-hep1	EV	Electroporation	Tumor growth inhibition	[129]
	mRNA	PTEN	MEF and DC	Exosome	Nanaoporation	Tumor growth inhibition	[130]
		UPRT	HEK293	Microvesicle	Co-culture	Tumor growth inhibition	[131]
	Anti-sense	STAT6	HEK293/M2 macrophage	Exosome	Co-incubation	Tumor growth inhibition	[132]
Gene editing tool	CRISPR-Cas9	CRISPR-PARP1	SKOV3	Exosome	Electroporation	Induce apoptosis	[133]
		CRISPR-WNT10B	HEK293	EV	Ultra-sonication	Tumor growth inhibition	[134]
Protein		Transferrin receptor binding peptide	MDA-MB-231	Exosome	Co-incubation	Tumor growth inhibition	[126]
		Tlyp-1	M1 macrophage	Exosome	Co-incubation	Tumor growth inhibition	[135]
		CD63/EGFR	M1 macrophage	EV	Electroporation	Tumor growth inhibition	[136]
Combination		CPPO/Ce6/Dox-EMCH	THLG-HEK293	EV	Electroporation	Induce drug sensitivity	[137]
		Dox/Cho-miR-159	THP-15	Exosome	Co-incubation	Tumor growth inhibition	[138]
		5-FU/miR21	HEK293	Exosome	Co-incubation	Tumor growth inhibition	[139]
		CPT-SS-PR104A	HEK293	Exosome	Co-incubation	Tumor growth inhibition	[114]
		siGPX4/Fe3O4	Tumor cell	Apoptotic body	Active loading	Tumor growth inhibition	[140]

**Table 4 ijms-24-14960-t004:** Clinical trials of mTOR inhibitors against glioblastoma.

Drugs	Registration No.	Stage	Disease Type	Target	Status
Afatinib Dasatinib Palbociclib Everolimus Olaparib	NCT05432518	Early Phase I	Glioblastoma Recurrent disease Recurrent glioblastoma	mTOR and Tyrosine kinase	Not yet recruiting
AZD2014	NCT02619864	I	Glioblastoma multiforme	mTOR	Completed
AZD8055	NCT01316809	I	Glioblastoma multiform Anaplastic astrocytoma Anaplastic oligodendroglioma Malignant glioma Brain stem glioma	mTOR	Completed
XL765 (SAR245409)	NCT00704080	I	Mixed gliomas Malignant gliomas Glioblastoma multiforme	Dual PI3K and mTOR	Completed
Everolimus Temozolomide	NCT00387400	I	Brain and central nervous system Tumors	mTOR	Completed
XL765 (SAR245409) XL147 (SAR245408)	NCT01240460	I	Glioma Glioblastoma Astrocytoma grade IV	Dual PI3K and mTOR	Completed
CC-115	NCT01353625	I	Glioblastoma multiforme	Dual pan-PI3K and mTOR	Completed
DEC-205/NY-ESO-1 fusion protein CDX-1401 Sirolimus	NCT01522820	I	Glioblastoma Anaplastic astrocytoma	mTOR	Completed
GDC-0084	NCT03696355	I	Brain and central nervous system Tumors	Dual PI3K and mTOR	Active, Not recruiting
RMC-5552	NCT05557292	I	Glioblastoma Recurrent glioblastoma	mTOR	Not yet recruiting
MLN0128	NCT02142803	I	Adult glioblastoma	mTOR	Active, Not recruiting
Perifosine Temsirolimus	NCT02238496	I	Brain tumor, Recurrent glioblastoma Anaplastic astrocytoma Anaplastic oligodendroglioma Mixed glioma	Dual Akt and mTOR	Active, Not recruiting
PQR309	NCT02850744	II	Glioblastoma multiforme	Dual pan-PI3K and mTOR	Terminated
Everolimus	NCT00515086	II	Glioblastoma multiforme	mTOR	Terminated
CC-223	NCT01177397	I/II	Multiple myeloma Diffuse large B-Cell lymphoma Glioblastoma multiforme Hepatocellular carcinoma Non-small cell lung cancer Neuroendocrine tumors of non-pancreatic origin Hormone receptor-positive breast cancer	Dual PI3K and mTOR	Completed

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
