# Peer review of "Unveiling Novel Avenues in mTOR-Targeted Therapeutics: Advancements in Glioblastoma Treatment"

_ijms, 2023, doi:10.3390/ijms241914960_

Round 1

Reviewer 1 Report

The mTOR signaling plays a prime role in glioblastoma development and growth. Inhibiting mTOR shows significant improvement in treating glioblastoma, but challenges like off-target effects and drug resistance must be addressed through novel strategies for patients with this aggressive brain cancer. Here, the authors summarize the mTOR signaling targets, strategies for developing targeted mTOR inhibitors, optimized drug delivery systems, and personalized treatment approaches to minimize difficulties linked with mTOR inhibitors. Though the current version of this article is well-structured, the below-mentioned points need to be addressed prior to publication.

1. Similar to Table 1, it would enhance comprehension if the authors incorporate a tabular overview summarizing the currently available mTOR inhibitors (Section 3.0 mTOR Inhibitors and Their Limitations) and various therapeutic approaches (Sections 4.1 Combination Therapies, 4.2 Personalized Medicines, 4.3 Nanotechnology-based Drug Delivery, and 4.4 Extracellular Vesicles as Drug Delivery Vehicles). This table should summarize the drugs and their respective molecular signaling for inhibiting mTOR.

2. The section titled '5. Potential Biomarkers for mTOR Inhibitor Response' should be repositioned to precede '4.0. Strategies for Targeting mTOR in Glioblastoma'.

Requires some correction of grammatical and typographical errors.

Author Response

Response to the reviewers’ comments

Rewier#1 Comments and Suggestions for Authors

The mTOR signaling plays a prime role in glioblastoma development and growth. Inhibiting mTOR shows significant improvement in treating glioblastoma, but challenges like off-target effects and drug resistance must be addressed through novel strategies for patients with this aggressive brain cancer. Here, the authors summarize the mTOR signaling targets, strategies for developing targeted mTOR inhibitors, optimized drug delivery systems, and personalized treatment approaches to minimize difficulties linked with mTOR inhibitors. Though the current version of this article is well-structured, the below-mentioned points need to be addressed prior to publication.

  1. Similar to Table 1, it would enhance comprehension if the authors incorporate a tabular overview summarizing the currently available mTOR inhibitors (Section 3.0 mTOR Inhibitors and Their Limitations) and various therapeutic approaches (Sections 4.1 Combination Therapies, 4.2 Personalized Medicines, 4.3 Nanotechnology-based Drug Delivery, and 4.4 Extracellular Vesicles as Drug Delivery Vehicles). This table should summarize the drugs and their respective molecular signaling for inhibiting mTOR.

            Response: Thank you for the thoughtful critique. The suggested tables (Table 1: mTOR inhibitors and their targeted pathways (page: 5-6); Table 2: mTOR inhibitor-encapsulated nanocarriers as therapeutic agents and effects on tumor growth (page: 10-11); Table 3: Extracellular vesicles as delivery vehicles for therapeutic agents and their effects (page: 12-13) have been included into the appropriate sections in the revised manuscript.

  1. The section titled '5. Potential Biomarkers for mTOR Inhibitor Response' should be repositioned to precede '4.0. Strategies for Targeting mTOR in Glioblastoma'.

Response: Thank you to the reviewer for bringing it to our attention. We repositioned the sections as suggested and emphasize the importance of the corresponding sections in the revised manuscript. Please see page: 6-7.

Comments on the Quality of English Language

Requires some correction of grammatical and typographical errors.

Response: The English language in the manuscript has been edited thoroughly, the grammatical and typographical errors has been rectified, and some sections has been revised to address the concerns.

Reviewer 2 Report

Here the authors present a review article about the role of mTOR signaling pathway in glioblastoma and the potential of targeting mTOR as a therapeutic strategy. The authors discuss the molecular mechanisms of mTOR dysregulation in glioblastoma, the current status of mTOR inhibitors in preclinical and clinical studies, and the emerging therapeutic strategies to optimize mTOR-targeted treatments. They highlight the challenges and opportunities in developing effective and personalized mTOR inhibitors for glioblastoma management.

Major points:

The review provides a comprehensive overview of the mTOR signaling pathway and its involvement in glioblastoma pathogenesis and progression.

The authors evaluate the efficacy and limitations of various mTOR inhibitors and combination therapies in glioblastoma treatment.

They suggest new therapeutic avenues and potential advancements in glioblastoma treatment through targeting the mTOR pathway²[2].

Minor points:

The review lacks a clear structure and organization of the main sections and sub-sections.

The article does not provide a clear conclusion or future directions for further research.

 I would add more information on mTOR C1, or change the title to "Unveiling Novel Avenues in mTOR C1-Targeted Therapeutics: Advancements in Glioblastoma Treatment"

fine; some minor typos

Author Response

Reviewer#2 Comments and Suggestions for Authors

Here the authors present a review article about the role of mTOR signaling pathway in glioblastoma and the potential of targeting mTOR as a therapeutic strategy. The authors discuss the molecular mechanisms of mTOR dysregulation in glioblastoma, the current status of mTOR inhibitors in preclinical and clinical studies, and the emerging therapeutic strategies to optimize mTOR-targeted treatments. They highlight the challenges and opportunities in developing effective and personalized mTOR inhibitors for glioblastoma management.

Major points:

The review provides a comprehensive overview of the mTOR signaling pathway and its involvement in glioblastoma pathogenesis and progression.

The authors evaluate the efficacy and limitations of various mTOR inhibitors and combination therapies in glioblastoma treatment.

They suggest new therapeutic avenues and potential advancements in glioblastoma treatment through targeting the mTOR pathway²[2].

Minor points:

The review lacks a clear structure and organization of the main sections and sub-sections.

 Response: Thank you for the thoughtful critique; however, this contrasts greatly with Reviewer #1’s assessment: “the current version of this article is well-structured”. In response, we have reorganized a few sections in the revised manuscript to clearly convey the content of our manuscript to the readers.

The article does not provide a clear conclusion or future directions for further research. Done

Response: We highly appreciate the reviewer's concern and included a comprehensive figure (Figure 3) provides clarity on the future directions of our proposed approach for targeting mTOR signaling and its interconnected pathways in the context of glioblastoma treatment. In addition, rewrite the conclusion to communicate the clear future directions of the study.

I would add more information on mTOR C1, or change the title to "Unveiling Novel Avenues in mTOR C1-Targeted Therapeutics: Advancements in Glioblastoma Treatment" 

Response: Thank you for the constructive suggestion. We have highlighted the sections that describe the role and importance of mTORC1 in the revised manuscript. Currently, we believe that further additions of mTORC1 information may not be pertinent to the current manuscript.

Comments on the Quality of English Language

fine; some minor typos

Response: Thank you. The grammatical and typographical errors have been rectified.
